# Root and Shoot Response to Nickel in Hyperaccumulator and Non-Hyperaccumulator Species

**DOI:** 10.3390/plants10030508

**Published:** 2021-03-09

**Authors:** Stefano Rosatto, Mauro Mariotti, Sara Romeo, Enrica Roccotiello

**Affiliations:** Laboratory of Plant Biology, DISTAV-Department of Earth, Environment and Life Sciences, University of Genoa, Corso Europa 26, 16132 Genova, Italy; stefano.rosatto@gmail.com (S.R.); M.Mariotti@unige.it (M.M.); sara.romeo.genova@gmail.com (S.R.)

**Keywords:** dose-response, ecophysiology, facultative hyperaccumulator, metal uptake

## Abstract

The soil–root interface is the micro-ecosystem where roots uptake metals. However, less than 10% of hyperaccumulators’ rhizosphere has been examined. The present study evaluated the root and shoot response to nickel in hyperaccumulator and non-hyperaccumulator species, through the analysis of root surface and biomass and the ecophysiological response of the related aboveground biomass. Ni-hyperaccumulators *Alyssoides utriculata* (L.) Medik. and *Noccaea caerulescens* (J. Presl and C. Presl) F.K. Mey. and non-hyperaccumulators *Alyssum montanum* L. and *Thlaspi arvense* L. were grown in pot on Ni-spiked soil (0–1000 mg Ni kg^−1^, total). Development of root surfaces was analysed with ImageJ; fresh and dry root biomass was determined. Photosynthetic efficiency was performed by analysing the fluorescence of chlorophyll a to estimate the plants’ physiological conditions at the end of the treatment. Hyperaccumulators did not show a Ni-dependent decrease in root surfaces and biomass (except Ni 1000 mg kg^−1^ for *N. caerulescens*). The non-hyperaccumulator *A. montanum* suffers metal stress which threatens plant development, while the excluder *T. arvense* exhibits a positive ecophysiological response to Ni. The analysis of the root system, as a component of the rhizosphere, help to clarify the response to soil nickel and plant development under metal stress for bioremediation purposes.

## 1. Introduction

The root system provides the structural element of the rhizospheric microenvironment [1] determining a plant’s access to soil-borne elements [2]. The root surface plays a significant role in element uptake through membrane transporters and in some hyperaccumulators the root grows towards trace elements in soil [2]. This metallophilic behaviour allows plants to mainly develop roots towards metal-rich patches [3]. The induction of root proliferation (i.e., root foraging) in response to Ni, Cd and Zn in soils were reported in few hyperaccumulators, like *Noccaea caerulescens* (J. Presl and C. Presl) F.K. Mey., *Thlaspi goesingense* Halácsy, *Sedum alfredii* Hance and *Streptanthus polygaloides* Gray. [4,5,6,7]. Nonetheless, other hyperaccumulators do not show the same positive chemotropism towards metal-spiked soil [8]. A recent study shows that the polymetallic hyperaccumulator of Zn, Cd and Pb *Noccaea rotundifolia* (L.) Moench ssp. *cepaeifolia* has a larger root and shoot biomass in soils where there is a heterogeneous distribution of the metal in the growth substrate, with a prevalence of the avoidance strategy against metal-rich patches in the soil [9].

Depth and root morphology are also important traits in relation to uptake, although little is known about the relationship between root morphology and metal accumulation [2]. Many hyperaccumulators have been described as shallow-rooted (<0.5 m width) and with a high proportion of fine roots for the accumulation of elements [5,10], but deep-rooted herbaceous species (2 m width) exist [11] and the roots of many arboreal hyperaccumulator species have not yet been examined [2].

At the root surface, specific membrane transporters provide metal uptake sites for soil metals such as Ni [12] which bind metal chelators; some of which can facilitate root-to-shoot translocation or be involved in the metal tolerance [2,13].

Among natural metalliferous soils, serpentine soils have nutrient deficiency and a toxic concentration of metals such as Ni which usually ranges from 500 to 8000 (bioavailable Ni: 7 to >100 mg kg^−1^) [14,15,16,17,18,19,20]). Worldwide, researchers use the term “serpentine” to define abiotic factors such as rocks, soils, but also biotic components such as vegetation and other biota associated with ultramafic outcrops [21]. Serpentine soils provide particularly harsh and hostile conditions for most plant species [22], except for some endemic and threatened species [23] and the presence of some tolerant hyperaccumulator species [24,25,26,27]. Hyperaccumulation is most probably a defense strategy against pathogens [28] or competitors [29] that allows plant species to thrive on harsh serpentine soils [22], enabling them to accumulate more than 1000 mg kg^−1^ dry weight of metals such as Ni, Co, Cu, Pb, Zn, Mn in their aboveground tissues [30].

This “serpentine factor” as mentioned by [31] is caused by peculiar edaphic conditions such as the lack of nutrients (N, P, S, K, Na, Ca) and the high concentration of phytotoxic elements (Ni, Fe, Cu, Co, Cr, etc., [32], wide temperature ranges, occurrence of thin soil layer and consequently low organic content, combined with high surface runoff [24]. Due to their mineralogical and chemical properties, serpentinitic soils have often been overexploited to extract metals from their ultramafic rocks [33] causing a serious threat to the surrounding ecosystem. In particular, mining activity is responsible for soil degradation and groundwater pollution [34], due to metal leaching and active acid mine drainage AMD [35]. Although root depth, morphology and the preferential metal allocation in roots may partially explain the high concentration of trace elements in some species of hyperaccumulator [2], the relation between the root system and the metal accumulation need to be further examined. Furthermore, metal stress affects the light phase of photosynthesis with negative impacts on the performance of the photosystems [36]. Recent reports on plant physiological activity under metal stress [36,37,38,39,40] highlight adverse effects on photosynthesis caused by high concentrations of metal. However, few detailed studies investigated hyperaccumulators ecophysiological response to metal stress [41,42].

A previous study [43] suggests that the rhizospheric micro ecosystem complex may be a fundamental model for better understanding the dynamics of plant development and the Ni uptake for soil bioremediation purposes.

Therefore, can we assume a nickelophilic root development and nickel foraging and related shoot response in nickel-hyperaccumulators? To evaluate these traits, the main aim of the study was to assess possible alterations in the development of the root system (morpho-functional response to Ni) in terms of biomass and surface area and how this behaviour could affect plant ecophysiology in terms of photosynthetic performance.

## 2. Results

### 2.1. Evaluation of Root Area, Biomass, and Plant Water Content

The red colour intensity of the leaves shown by the 1% colorimetric dimethylglyoxime test (DMG) test enhances with increasing Ni concentration, suggesting a nickel accumulation in the leaf epidermis for both the hyperaccumulator species starting from 100 mg kg^−1^ of Ni concentration in pot.

The qualitative observation of the roots of the test species scanned with the ImageJ software does not clarify any relationship between the root surface area and the increasing concentrations of Ni from 0 mg kg^−1^ to 1000 mg kg^−1^ (Appendix A). However, the analysis in non-accumulator *T. arvense* (Figure 1D and Figure 2D) show an increase in terms of root surface area and root dry weight (*p* < 0.01) at increasing Ni concentrations, which is less evident for the aboveground organs (*p* < 0.05). It is interesting to note the significant increase in root (*p* < 0.01) and shoots (*p* < 0.01) biomass of the facultative Ni-hyperaccumulator *A. utriculata* (Figure 2A), although this is not supported by an increase of the root surface (Figure 1A).

*N. caerulescens* and *A. montanum* do not exhibit significant differences in terms of root surface area and biomass, except for *N. caerulescens* at Ni 1000 (Figure 1B,C and Figure 2B,C).

In the facultative Ni-hyperaccumulator *A. utriculata*, the Ni concentration is positively correlated with fresh and dry shoot biomass (*p* < 0.001) and with the root dry biomass (*p* < 0.01). In *N. caerulescens* Ni seems to positively affects the root water content (*p* < 0.01) (Table 1).

Non-hyperaccumulator species behave differently: while in *A. montanum* the presence of Ni does not seem to affect the considered biological parameters, *T. arvense* shows a positive correlation between Ni and the root surface area and biomass, the root: shoot biomass ratio, the water content of the aerial organs (*p* < 0.001) and the shoot dry weight (*p* < 0.01) (Table 1).

A clear difference in terms of Ni uptake between the hyperaccumulator species exists, as indicated in Table 2. At the maximum Ni concentration, the facultative hyperaccumulator *A. utriculata* accumulates ~1000 mg Ni kg^−1^ in the aboveground biomass, while *N. caerulescens* can accumulate seven times higher Ni concentration (6798 mg Ni kg^−1^) compared to the other species.

In the non-hyperaccumulator species, the shoot metal uptake at Ni 1000 is significantly lower compared hyperaccumulators and the range varies between 100 and 200 mg kg^−1^.

### 2.2. Photosynthetic Efficiency

The curves in Figure 3 are plotted on a logarithmic axis to observe the fluorescence over time up to the maximum fluorescence value. On the other hand, spider (or radar) plots (Figure 4) provide a method of comparing the spread of individual parameter values for selected records within a data set and a visualisation of which parameters have greater sensitivity to certain types of stress.

Figure 3 summarizes the transient fluorescence analysis for controls and treatments at Ni 1000 mg kg^−1^. The curves represent the average values of samples measured for the indicated species at Ni 0 (light colour) and Ni 1000 mg kg^−1^ (dark colour). The peaks represent the fluorescence values measured at 50 µs (0), 2 ms (J), 30 ms (I) and maximal (P) [44].

Through the evaluation of the Fv/Fm ratio, the Photosystem II efficiency was assessed for the test species. Instead, P.I. is an index of the vitality of the sample: it expresses the potential capacity for energy conservation.

Only *N. caerulescens* shows a greater Fm in the control than at the maximum Ni concentration, while *A. utriculata* and *A. montanum* exhibit the highest values of Fm.

Figure 4 represents a set of spider plots showing the differences between the test species at increasing Ni concentration in terms of recorded parameters.

Each axis of the plot corresponded to one of the seven parameters (F0, Fv, Fm, Fv/Fm, Tfm, Area and P.I.) and the average value of each parameter for the control species was used as the comparative value.

In *A. utriculata* (Figure 4A), Tfm, Area and P.I. increase in treatment (*p* < 0.001, *p* < 0.001 and *p* < 0.05 respectively). The parameters do not seem to differ considerably in *N. caerulescens*, while in non-hyperaccumulators there is a reduction in the P.I. except for Ni 50 mg kg^−1^ (*A. montanum,*
Figure 4C) and Ni 1000 mg kg^−1^ (both species, Figure 4C,D). Clear differences are observed between the Ni treatments in non-accumulators. In particular, Tfm and P.I. reach the maximum value at Ni 1000 mg kg^−1^.

Data summarized in Table 3 show that in *A. utriculata*, Performance Index (P.I., *p* < 0.05) and the time to reaching Fm (Tfm, *p* < 0.001) increase at increasing Ni concentrations. Similarly, the latter enhances (*p* < 0.001) in *N. caerulescens*, although the maximum fluorescence (Fm) decrease (*p* < 0.01), and there is no significant difference in terms of P.I. In *A. montanum* a positive correlation between dark level fluorescence (F0) and Ni (*p* < 0.001) and between the latter and Fm (*p* < 0.05) is observed. On the other hand, in the same conditions, *T. arvense* exhibits a sensible reduction of F0 (*p* < 0.01), but a highly significant positive correlation (*p* < 0.001) between Ni and the maximum efficiency of photosystem II (Fv/Fm), the P.I. and the Tfm.

## 3. Discussion

### 3.1. Evaluation of Root Area, Biomass, and Plant Water Content

The root surface and biomass development highlighted that at increasing available Ni concentration, both the polymetallic hyperaccumulator of Ni, Cd and Zn *N. caerulescens* [45,46] and the Ni-facultative hyperaccumulator *A. utriculata* [47] did not exhibit a clear dose-response effect by Ni as demonstrated by Whiting et al. in *N. caerulescens* [48]. The same response has been already highlighted for the hyperaccumulator *N. caerulescens* grown even at 100 μM Ni concentration without significant alterations of root dry weight [49].

Efficient sequestration and detoxification and high metal mobility in the root tissues in hyperaccumulators were well-known phenomena [2,49,50]. In addition, the root surfaces of hyperaccumulator species in response to Ni, even at a different level of affinity (higher for obligate hyperaccumulator and lower for facultative hyperaccumulator), supported their specific root response to metal, as demonstrated for Zn [51].

The response of the root system is interesting with respect to the tested concentrations, that seems to be comparable with natural serpentine soils where bioavailable Ni ranges from 7 to >100 mg kg^−1^ [15,18,19], while total Ni ranges between 500 and 8000 mg kg^−1^ [14,16,17].

The effects of metal concentration on plant growth and biomass are often used to assess phytotoxicity in non-hyperaccumulator species [52] or to evaluate differences in trace element concentration from inoculated and uninoculated plants [53,54].

In the pot test the non-hyperaccumulator *A. montanum* shows constant root area and plant biomass. Surprisingly, root biomass and root area of *T. arvense* significantly increase with Ni treatments (but 2.5 times less than hyperaccumulators) despite other studies showing a disrupted growth in non-accumulator plants, i.e., *Lactuca sativa* L. [25], or severe root biomass reduction in non-accumulator *Cicer arietinum* L. [8]. A possible explanation is that the pot test lasted long time causing a sort of adaptation of the involved species, but further investigations will clarify this point.

At the highest Ni concentration in pots, both hyperaccumulator species gave a positive reaction to the DMG test that highlighted an evident Ni translocation to aboveground biomass as expected [55]. In addition, *A. utriculata* significantly increased aboveground biomass while *N. caerulescens* was comparable to the control, as indicated by Pollard et al. [24] and similar to what was reported by Moradi et al. [8] for *Berkheya coddii* Roessler.

The phenotypic plasticity of the facultative hyperaccumulator species *A. utriculata* may depend on epigenetic modifications enabling a rapid adaptation to metal stress [56,57,58] or adaptation to edaphic conditions [59]. This may lead to a greater development of *A. utriculata* in the aboveground biomass where Ni accumulates. The significant increase of root biomass at increasing Ni treatment was not yet appreciated, even if *A. utriculata* had no negative effects on biomass, development and plant water content under Ni stress [60]. This aspect can probably be related to the species ability to concentrate metals in its aboveground biomass without display toxicity symptoms [60].

Considering that Ni-hyperaccumulators sequester Ni in shoots and by contrast Ni-excluders store Ni in roots, we could hypothesize in agreement with Seregin et al. [49] that roots systems of both hyperaccumulators *A. utriculata* and *N. caerulescens* do not show significant differences in terms of root surface and biomass at increasing Ni (except *N. caerulescens,* which suffered Ni 1000 mg kg^−1^) because they are more able to tolerate the metal stress while *T. arvense* specifically act as an excluder, increasing root area and biomass to sequester metal (phytostabilisation).

### 3.2. Photosynthetic Efficiency

Consistent with the biomass results, and consistent with other plants able to accumulate metals [61,62], the maximum primary photochemical efficiency of PSII, Fv/Fm, and the performance index (PI) [63] do not differ in the hyperaccumulator species at increasing Ni concentrations, while it significantly declines in *A. montanum.* This could be due to the great sensitivity to the fluctuations of environmental abiotic factors [64], such as Ni stress. Both parameters increase in *T. arvense,* although Fm is lower than the other species. A low Tfm may indicate sample stress [63] which causes the Fm to be achieved much earlier than expected, similar to what happens in *A. montanum.*

However, as confirmed by some studies [65,66], our results suggest that photosynthetic efficiency is less sensitive than the morpho-functional responses to Ni in terms of biomass and surface. The maximum efficiency of PSII (Fv/Fm) appears to be strictly dependent on the plant species tested, since the use of tolerant species might reduce their sensitivity to metal stress in the soil. Further studies on different species will be needed to clarify these aspects.

A comprehensive explanation could concern a Ni adaptation of *A. utriculata* to Ni input during the plant growth due to a possible phenotypic plasticity [67]. The same will probably occur to *T. arvense*, specifically at the root level where root surface and biomass increase to cope with a high level of Ni. However, epigenetic aspects involved in plant adaptation to strongly stressed environments and detoxification mechanisms need to be verified by further investigations [68].

In addition, the previous study [43] demonstrates that bacterial and fungal microbial communities isolated from the rhizosphere of hyperaccumulator species are subjected to selective pressure due to the ‘rhizophere effect’ [69,70,71,72,73,74] and to the high soil metal content [75,76,77,78,79]. Indeed, bacteria and fungi mainly thrive in rhizospheric serpentinitic soils compared to bare and non-metalliferous ones [43].

On the basis of plant growth-promoting (PGP) traits of many microorganisms [43,80,81,82,83,84,85,86,87], we can hypothesize the key role of the culturable rhizospheric microbiota to ameliorate the performance of metal hyperaccumulator species for soil remediation purposes.

Future research will better clarify the mutual behaviour of the rhizobiota associated with the root system of hyperaccumulators in the future perspective of the development of a holistic bioremediation system (plant–bacteria–fungi).

## 4. Materials and Methods

### 4.1. Plant Species

*Alyssoides utriculata* (Ni in leaves 36–2236 mg kg^−1^ DW, Figure 5A) is an evergreen shrub with good biomass. It is a Ni facultative hyperaccumulator [18,35] occurring on both metalliferous and ‘normal’ soils. The species ranges primarily in the north-eastern Mediterranean region [88]. The greater abundance of *A. utriculata* in low-competition serpentine soils compared to adjacent non-serpentine sites, suggests preadaptation tolerance traits [18]. It is able to accumulate greater quantities of Ni in the aboveground biomass even in soils with low metal level, compared to the typical serpentine non-hyperaccumulator species [60]. Despite the medium-high ability to concentrate Ni in shoots, *A. utriculata* species is of a key importance because it is a native Mediterranean hyperaccumulator that can be exploited for improved phytoremediation purposes in this climate.

*N. caerulescens* (Ni in the shoots 1000–30000 mg kg^−1^ DW, Figure 5B) is a herbaceous biennial plant, found in Europe and in the USA [46]. It has been studied extensively for its ability to hyperaccumulate several metals [46]. Some populations of the genus *Noccaea* (syn. *Thlaspi*) hyperaccumulate Ni in the serpentinitic soil, whereas other populations are capable to uptake Zn and Cd [89], suggesting that hyperaccumulation is monophyletic [90]. *N. caerulescens* also known as ‘montane crucifer’ [91] or ‘alpine pennycress’ [92] includes populations that differ in morphological and physiological characteristics, exhibiting a wide range of accumulation and metal tolerance [93]. In Europe, *N. caerulescens* shows three ecological groups that correspond to three edaphic environments [94]. Two ecological groups are typical of metalliferous soils: the Calamine group develops in soils rich in Zn, Cd and Pb, while the Serpentine group is characterized by populations that thrive in Ni-rich soils derived from serpentinite ultramafic rocks. Finally, the third group includes non-metalliferous populations [95].

*Alyssum montanum* L. (Figure 5C) and *Thlaspi arvense* L. (Figure 5D) were the related non-hyperaccumulator species used for comparison, as in other experiments [45,46,96,97,98].

*A. montanum*, the ‘mountain gold’ [97] is a perennial herb belonging to the Brassicaceae family [99]. It is known to be a monophyletic polyploid complex [100] diversified in the last 2 million years [101] and characterized by a wide geographical and ecological range including more than 30 species and subspecies [99]. It is located in Europe, western Asia and northern Africa, with a clear species diversity in the Mediterranean area [102,103,104,105].

*Thlaspi arvense* L. (field pennycress, pennycress herein [106] is an oilseed-producing plant [107] member of Thlaspide, a tribe of Brassicaceae [108]. Pennycress is a winter annual with a short life cycle [109] native to Eurasia and it is widespread in temperate regions of the northern hemisphere [110]. It is closely related to the model *Arabidopsis thaliana* (L.) Heynh. [111]. Pennycress exhibits winter growth and extreme cold tolerance traits [112] and it is known to provide ecosystem services in the cold season such as habitat and food source for pollinator insects, weed suppression, and reduction of soil erosion and nutrients leaching [109,113,114,115,116].

### 4.2. Seed Collection

Hyperaccumulator plants *A. utriculata* and *N. caerulescens* (Figure 5A,B) were grown from seeds collected according to international guidelines (ENSCONET, 2009), from serpentine soils in Liguria (NW Italy) in July 2016. Samples were harvested from the eastern Ligurian Alps (Voltri Massif, 44°28′49″ N, 8°40′44″ E). The presence of Ni in the mother plants was assessed by means of a colorimetric field dimethylglyoxime test [117,118]. All plants yielded a dimethylglyoxime-positive reaction.

The related non-hyperaccumulator species *A. montanum* L. (Figure 5C) and *T. arvense* L. (Figure 5D) were collected from ‘normal’ soil and used for comparison. Seeds were provided by herbarium specimens: *A. montanum* from the Jardin Botanique de Bordeaux FR0BORD120310-Causse–Méjean, and *T. arvense* were provided by the Botanischer Garten Ulm IPEN XX-0-ULM-1998-F-152.

### 4.3. Pot Experiment

#### 4.3.1. Evaluation of Root Area, Biomass, and Plant Water Content

The peat–sand mix (2:1) was chosen as a growing substrate; the substrate was sterilized at 120 °C for 20′, and oven dried at 60 °C. The final pH of substrate was measured by mixing an aliquot of soil with deionized water (ratio 1:3), and to obtain a pH of 6–6.5, slaked lime with Ca and Mg was added to the dry soil. After the assessment of pH, the soil water content at field capacity on a volume basis [119] was assessed to calculate the water holding capacity (WHC): 100 mL of water was added to 100 mL of dry soil placed in a funnel on a graduated cylinder. After waiting at least 1 h until the last drop, the WHC (%) was calculated based on the volume of water retained by the soil. Finally, the soil was transferred to 10 cm Ø pot.

To evaluate the root surface response to increasing concentration of available Ni, soil was homogeneously hydrated with a 70% WHC solution of 1/4 strength Hoagland’s basal salt mixture n.2 (Sigma-Aldrich, St. Louis, MO, USA) and metallic salt (NiSO_4_*6H_2_O) [4,8] was added to obtain increasing concentrations of available Ni: 0, 50, 100, 200, 500, 1000 mg L^−1^ respectively. Afterwards, seeds were surface-sterilized with sodium hypochlorite (NaClO) 10% for 10′ [120] and placed in pot (one seed per box, five replicates each concentration).

Pots were transferred to a greenhouse and the plants were grown in semi-natural conditions at controlled temperature (T = 19–22 °C) for 120 days, replenishing the plants with deionized water and monitoring plant growth two times a week.

During the third month of growth, the water-soluble fertilizer Leader N-P-K (20-10-20 + MgO + Me) was solubilised in deionised water at the concentration of 0.5 g L^−1^ and supplied for each pot once a week for one month.

At the end of the test, Ni accumulation in leaves was evaluated via a colorimetric dimethylglyoxime test (DMG 1%, Sigma-Aldrich, in ethanol 95%, [117] for each species and treatment. One mature leaf each plant was collected and placed in a solution of 1% DMG in ethanol 95°. Leaves turn red when a positive reaction occurs (high amount of Ni is stored in leaf epidermis).

Each plant was gently removed from the substrate, washed with tap water and then with deionised water, divided into root and shoot and weighed for fresh biomass. Roots were scanned and the resulting images were processed and analysed with ImageJ software to assess the root surface area. ImageJ is designed for scientific multidimensional images. It allows the user to create binary images (i.e., black and white), calibrate and analyse them, distinguish root from the background, and analyse the surfaces obtained to obtain the whole root area.

First, the measurement scale was set using a graduated scale in each scanned image. The root image was then converted to 8-bit images. Finally, images were converted into a black and white image by thresholding to evaluate the surface area.

Finally, DW were determined after oven-drying (60 °C, 48 h). Leaves were powdered using a ball mill (Retsch MM2000, Haan, Germany), preceding XRF analysis.

The chemical characterization of the dried samples was carried out on the granulometric fraction <2 mm by using an X-ray field portable spectrophotometer (X-MET7500 FP-EDXRF Analyser, Oxford Instruments, Abingdon-on-Thames, UK) that allows non-invasive and non-destructive analyses, providing information about chemical composition of the shoots. Quantitative analyses were obtained from trace level (ppm) to 100% for elements with atomic number ≥12 and the data quality level of the analyses was defined according to the Method 6200 of the U.S. Environmental Protection Agency [121]. This procedure was performed thanks to the collaboration with the Geospectra s.r.l, spin off of the University of Genoa. It represents an efficient, alternative approach to traditional laboratory analysis, allowing measurement of the concentration of a wide range of chemical elements.

#### 4.3.2. Photosynthethic Efficiency

Chlorophyll fluorescence is a simple and non-invasive measurement technique of photosystem activity (PSII) in which the light energy absorbed by chlorophyll is partly re-emitted as light [122].

To estimate the plant physiological condition at the end of the growth period, 10 measurements of photosynthetic efficiency in each plant were performed on leaves with the digital fluorimeter Handy-Pea (Hansatech Instruments, King’s Lynn, UK). It provides the high time resolution essential to perform measurements of fast chlorophyll fluorescence induction kinetics. Leaf samples were covered with the leaf clip which has a small shutter plate to keep it close when the clip is attached so that light is excluded, and dark adaptation takes place. After 20 min the chlorophyll fluorescence signal received by the sensor head during recording is digitised within the Handy PEA control unit using a fast Analogue/Digital converter.

The fluorescence transient is a tool to characterize and screen photosynthetic samples [63]. General parameters recorded by fluorimeter are:F0 is the minimum fluorescence value and represents emission by the excited chlorophyll *a* molecule in the antenna structure of Photosystem II.Fm is the maximum fluorescence value obtained after the application of a saturation pulse to the dark-adapted leaf.Fv is the variable fluorescence, and it denotes the variable component of the recording and relates to the maximum capacity for photochemical quenching.Fv/Fm is widely used to indicate the maximum quantum efficiency of Photosystem II. It is a sensitive indication of plant photosynthetic performance.Tfm is used to express the time at which the maximum fluorescence value (Fm) was reached.Area above the fluorescence curve between F0 and Fm is proportional to the pool size of electron acceptors.P.I. (Performance Index) is essentially an indicator of sample vitality.

The excitation light consisted of a 1 s pulse of ultra-bright continuous red radiation (650 nm peak wavelength), provided by an array of three light-emitting diodes focused on a leaf surface of 5 mm at an intensity of 3500 μmol photons per square meter (m^2^) per second (s). The analysis of the transient was based on the fluorescence values measured at 50 µs (F0), 2 ms, 30 ms, and maximal (Fm) after about 300 ms [60,63]. The fluorescence induction is well represented by the Kautsky induction curve [123].

The fluorescence emission provides information about the phytochemical efficiency, the heat dissipation and the electron transfer reactions [124,125,126,127]; therefore, the presence of any type of stress results in photoinhibition and a low Fv/Fm ratio [122]. Therefore, a quick screening of the photosynthetic efficiency shows the trend of growth and plant yield [128,129].

Data on photosynthetic efficiency and plant physiological performance, obtained from the averages of measurements on the test species, have been processed with PEA-Plus software (Hansatech Instruments).

### 4.4. Data Analysis

The statistical analyses were performed with Statistica 8.0 (Statsoft Inc., Tulsa, OK, USA) software.

The averages were presented with their standard deviations (SD). Non-parametric tests were used to avoid data transformation. Normality of parameters were evaluated with the Shapiro–Wilk test. Correlations between variables were analysed using Spearman’s correlation coefficient (ρ) using different level of significance (α: 0.05, 0.01, 0.001), since data exhibit a non-normal statistical distribution.

Moreover, the R/S ratio for both fresh and dry biomass and the water content (100 * DW/FW) in the root and shoot were evaluated [60].

The open-source software ImageJ ([130]; http://imagej.nih.gov/ij/ accessed October 2019) was used to determine the root surface on the whole roots of each species and treatment (*n* = 90).

## 5. Conclusions

The evaluation of the morpho-functional and ecophysiological alterations of hyperaccumulator and non-hyperaccumulator species under Ni stress allows us to better clarify the behaviour of plant species suitable for soil bioremediation at the root and shoot level.

We highlighted a nickelophilic behaviour and Ni foraging of the root system mainly in the facultative hyperaccumulator *A. utriculata* that may be due to its phenotypic plasticity combined with epigenetic aspects that make this species an excellent plant model for studying the Ni uptake. Non-hyperaccumulators show symptoms of metal stress or nickel avoiding strategies, as in the case of *T. arvense.* This latter increases both root surface and overall biomass. Perhaps its Ni-excluder traits could explain this response since metal is immobilized at the root level to avoid translocation to shoot biomass and possible related toxicity symptoms. However, further studies are required to better understand its ecophysiological behaviour under Ni stress.

This study provides the evidence for a careful selection of the best performing species for phytoremediation purposes considering root metallophilic behaviour and consequent better adaptation on metal-polluted soils.

## Figures and Tables

**Figure 1 plants-10-00508-f001:**
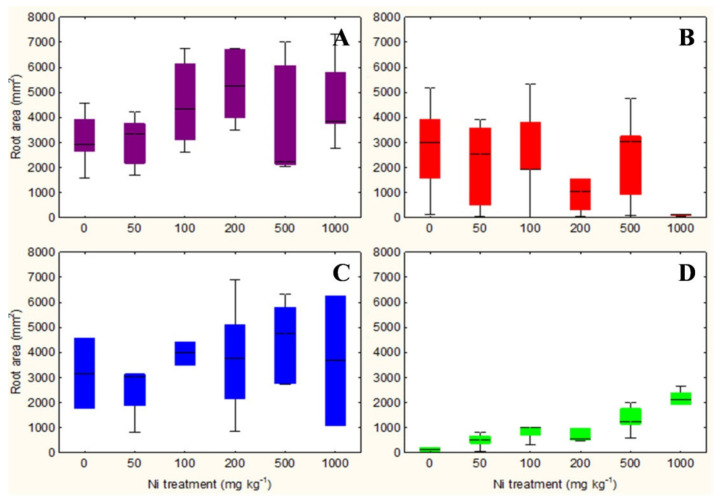
Box-and-whisker plots showing the root surface area (mm^2^) of test hyperaccumulator species at increasing Ni concentrations (0–1000 mg kg^−1^): (**A**) *A. utriculata*, (**B**) *N. caerulescens* and non-hyperaccumulator species (**C**) *A. montanum*, (**D**) *T. arvense.* In each box, the central line marks the median of the data; the box edges represent the first and third quartiles; whiskers show non-outlier range. *n* = 20 each treatment.

**Figure 2 plants-10-00508-f002:**
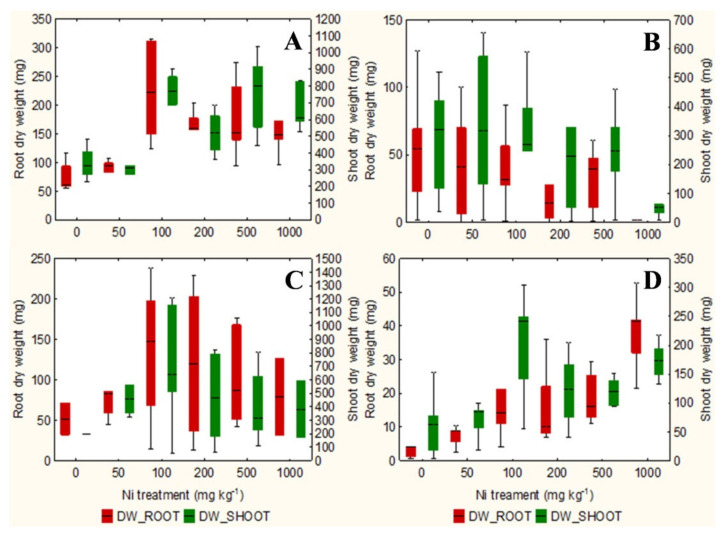
Box-and-whisker plots showing the root and shoot dry biomass (mg) of test hyperaccumulator species at increasing Ni concentrations (0–1000 mg kg^−1^): (**A**) *A. utriculata*, (**B**) *N. caerulescens* and non-hyperaccumulator species (**C**) *A. montanum,* (**D**) *T. arvense.* In each box, the central line marks the median of the data; the box edges represent the first and third quartiles; whisker show non-outlier range. *n* = 20 each treatment.

**Figure 3 plants-10-00508-f003:**
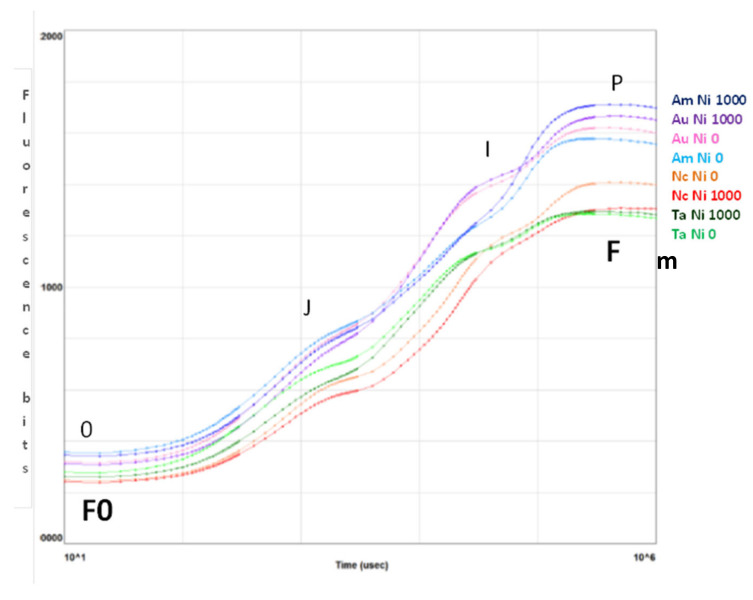
Fluorescence transient analysis of test species distinguished by colour of the curves (purple: *A. uticulata*, red: *N. caerulescens*, blue: *A. montanum*, green: *T. arvense*) at Ni 0 mg kg^−1^ (light colour) and Ni 1000 mg kg^−1^ (dark colour). The peaks are denoted by letters 0, J, I, P and correspond to fluorescence values measured at 50 ms (F0, step 0), 2 ms (step J), 30 ms (step I), and maximal (Fm, step P). Data are the mean of ten measurements per plant. *n* = 100 each species.

**Figure 4 plants-10-00508-f004:**
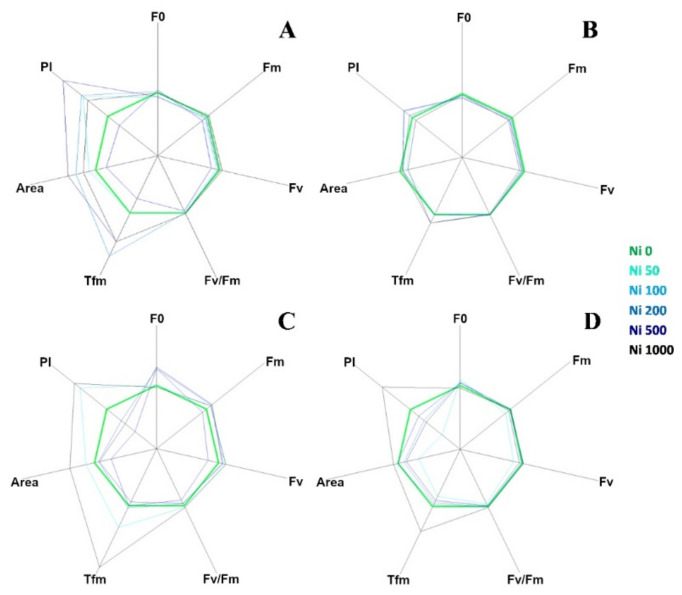
Fluorescence transient analysis of test species represented by spider plots: (**A**) *A. utriculata*, *(***B**) *N. caerulescens*, (**C**) *A. montanum*, (**D**) *T. arvense*. Parameters considered: F0: dark level fluorescence, Fm: maximum fluorescence, Fv: variable fluorescence, Fv/Fm: maximum efficiency of photosystem II (PSII), Tfm: time (ms) to reaching Fm, Area: area above the fluorescence curve between F0 and Fm, P.I. performance index. Increasing concentrations of nickel correspond to darker shades of blue; green identifies the control. Data are standardized for Ni 0 mg kg^−1^. Data are the mean of ten measurements per plant. *n* = 300 each species.

**Figure 5 plants-10-00508-f005:**
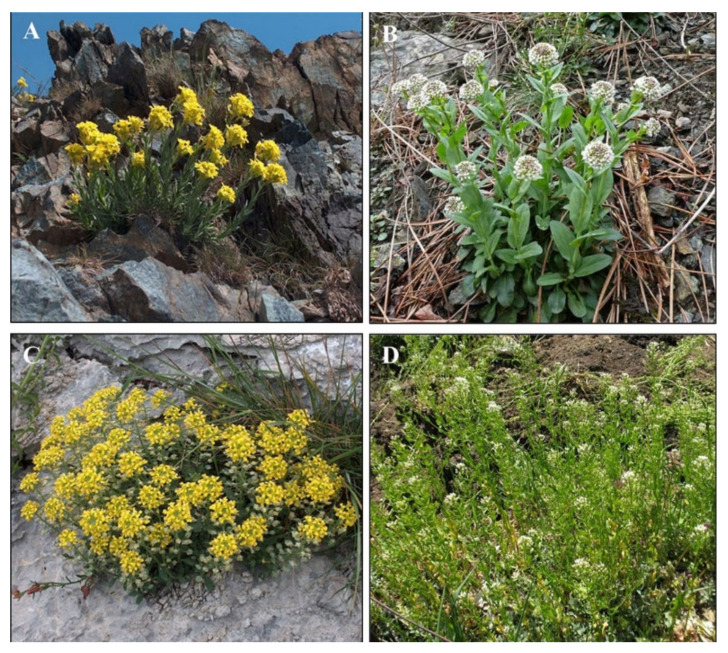
Test species: Ni-hyperaccumulator (**A**) *A. utriculata* and (**B**) *N. caerulescens* and non-hyperaccumulator (**C**) *A. montanum* and (**D**) *T. arvense,* photographs by S. Marsili (**A**,**C**), N. Mazzoni (**B**), G. Nicolella (**D**).

**Table 1 plants-10-00508-t001:** Spearman’s rank correlations coefficients between Ni concentration and biological parameters of test species: root surface area, fresh and dry weight (FW, DW) of root and shoot, root/shoot ratio (R/S) and water content (%). *n* = 20 each treatment. *Au: A. utriculata, Nc: N. caerulescens, Am: A. montanum, Ta: T. arvense.* ++ hyperaccumulator species, + non-hyperaccumulator species. * *p* < 0.05, ** *p* < 0.01, *** *p* < 0.001, NS: not significant.

Biological Parameters	Ni Treatment			
	*Au++*	*Nc++*	*Am+*	*Ta+*
Root surface area	NS	NS	NS	0.74 ***
FW_root	NS	NS	NS	0.66 ***
DW_root	0.53 **	NS	NS	0.78 ***
FW_shoot	0.61 ***	NS	NS	NS
DW_shoot	0.61 ***	NS	NS	0.48 **
R/S FW	NS	−0.42 *	NS	0.68 ***
R/S DW	NS	NS	NS	0.64 ***
% water R	NS	0.54 **	NS	NS
% water S	NS	NS	NS	0.60 ***

**Table 2 plants-10-00508-t002:** X-ray fluorescens chemical analysis of Ni concentration measured on shoots of test species *A. utriculata, N. caerulescens, A. montanum* and *T. arvense*. *n* = 10 each species. ++ hyperaccumulator species, + non-hyperaccumulator species.

Species	Ni Treatment (mg kg^−1^)	Ni Concentration (mg kg^−1^)
***A. utriculata* ++**	0	18.33 ± 2.00
	1000	999.67 ± 8.00
***N. caerulescens* ++**	0	76.67 ± 3.00
	1000	6798 ± 24.33
***A. montanum* +**	0	4.00 ± 1.33
	1000	127.33 ± 4.00
***T. arvense* +**	0	0.00
	1000	177.00 ± 5.00

**Table 3 plants-10-00508-t003:** Spearman’s rank correlations coefficients between Ni concentration and fluorescence parameters of test species (F0: dark level fluorescence, Fm: maximum fluorescence, Fv/Fm: maximum efficiency of PSII, Tfm: time (ms) to reaching Fm, P.I. Performance Index). *n* = 300 each species. *Au: A. utriculata, Nc: N. caerulescens, Am: A. montanum, Ta: T. arvense.* ++ hyperaccumulator species, + non-hyperaccumulator species. * *p* < 0.05, ** *p* < 0.01, *** *p* < 0.001, NS: Not Significant.

Parameters	Ni Treatment			
	*Au++*	*Nc++*	*Am+*	*Ta+*
F0	NS	NS	0.25 ***	−0.16 **
Fm	NS	−0.17 **	0.15 *	NS
Fv/Fm	NS	NS	−0.22 **	0.24 ***
Tfm (ms)	0.38 ***	0.37 ***	NS	0.25 ***
Area	0.30 ***	NS	NS	0.19 ***
P.I.	0.13 *	NS	−0.21 **	0.31 ***

## Data Availability

Not applicable.

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
