# Peer review of "Root and Shoot Response to Nickel in Hyperaccumulator and Non-Hyperaccumulator Species"

_plants, 2021, doi:10.3390/plants10030508_

Round 1
Reviewer 1 Report
In the present study, the authors evaluated and estimated “Root response to nickel in hyperaccumulator and non-hyperaccumulator species”. The study is interesting and the information is useful. The study of this manuscript is study that is a continuation of previously presented research these authors.
I would suggest minor modifications to further improve the clarity of the manuscript.
Comments to the Author:
- The authors did not present any hypothesis at the beginning
- correct Conclusion - make sure the conclusion is short and solid. Add a practical implications statement.
Author Response
In the present study, the authors evaluated and estimated “Root response to nickel in hyperaccumulator and non-hyperaccumulator species”. The study is interesting and the information is useful. The study of this manuscript is study that is a continuation of previously presented research these authors.
I would suggest minor modifications to further improve the clarity of the manuscript.
Thank you for the useful comments and suggestions to our manuscript. Detailed answers are listed below, point by point.
Comments to the Author:
- The authors did not present any hypothesis at the beginning
Thank you for the comment, we added a sentence on the scientific hypothesis in the Introduction section, “Can we therefore assume a nickelophilic root development and nickel foraging and related shoot response in nickel-hyperaccumulators?” - Lines 80-81
- correct Conclusion - make sure the conclusion is short and solid. Add a practical implications statement.
We have reorganized and shortened the Conclusions to make them more solid and we added the possible future practical implications of this research (lines 443-458).
Reviewer 2 Report
The present paper analyzed the response of 4 different plant species to a soil Nickel pollution. The results showed that Hyperaccumulators A. utriculata and N. caerulescens did not show a decrease in plant biomass at high level of Ni; while the non-hyperaccumulator A. montanum suffers the metal stress. A positive ecophysiological response was found for the excluder T. arvense. The topic is of interest and within the scope of Plants; covering the interest areas of plant anatomy and morphology; and plant physiology and ecophysiology. The assay was well conducted and planned, and the results and methodology were well explained with clear figures and tables. However, I would like to share several impressions with the authors.
The first thing is that the title is not enough consistent with the manuscript. The title only refers to roots, but more than the half of the figures and tables of the manuscript analyzed the shoots.
In Figures 1 and 2 it can be appreciated the low root values of the N. caerulescens at a Ni concentration of 1000 mg·Kg-1. If, for this treatment the plants died, then the sentence: “native metalliferous hyperaccumulator A. utriculata and N. caerulescens are not affected by increasing Ni concentrations at the root and shoot level - Line 426” Should be changed or at least nuanced/clarified.
In Table 2 all the values of Ni concentration should have the same format according for example to the number of significant figures.
In the discussion (Lines 192 and 200) so much emphasis is put in Zn but in the work this element has not been studied.
In Line 258, the acronym PGP should be changed for plant growth promoting, since was not described previously.
Since the root is being studied, Ni concentration levels of the root should have been measured, giving more strength to the ideas of: “highlight a Ni translocation - Line 216”, “do not show differences at increasing Ni - Line 230”, “T. arvense, specifically at the root level - Line 250” and “immobilized at the root level - Line 433”.
Author Response
The present paper analyzed the response of 4 different plant species to a soil Nickel pollution. The results showed that Hyperaccumulators A. utriculata and N. caerulescens did not show a decrease in plant biomass at high level of Ni; while the non-hyperaccumulator A. montanum suffers the metal stress. A positive ecophysiological response was found for the excluder T. arvense. The topic is of interest and within the scope of Plants; covering the interest areas of plant anatomy and morphology; and plant physiology and ecophysiology. The assay was well conducted and planned, and the results and methodology were well explained with clear figures and tables. However, I would like to share several impressions with the authors.
Thank you for the useful comments and suggestions to our manuscript. Detailed answers are listed below, point by point.
The first thing is that the title is not enough consistent with the manuscript. The title only refers to roots, but more than the half of the figures and tables of the manuscript analyzed the shoots.
Thank you, as suggested, title was modified to include the shoots.
In Figures 1 and 2 it can be appreciated the low root values of the N. caerulescens at a Ni concentration of 1000 mg·Kg-1. If, for this treatment the plants died, then the sentence: “native metalliferous hyperaccumulator A. utriculata and N. caerulescens are not affected by increasing Ni concentrations at the root and shoot level - Line 426” Should be changed or at least nuanced/clarified.
Although the highest tested concentration of Ni (1000 mg/kg) negatively influenced the development of N. caerulescens, the plants are still viable at the end of the treatment. For this reason, we specify in the abstract that “metal is immobilized at the root level to avoid translocation”
The conclusion section was shortened and reorganized according to what requested from reviewer 1.
In Table 2 all the values of Ni concentration should have the same format according for example to the number of significant figures.
Done, thank you.
In the discussion (Lines 192 and 200) so much emphasis is put in Zn but in the work this element has not been studied.
N. caerulescens is a known polymetallic hyperaccumulator with high affinity for several bivalent cation, including Zn. It provides a large root surface area that promotes metal uptake occurring at high metal concentration in soil. Future studies may clarify the linkage between metallic elements accumulated by plants. However, the reference to Zn accumulation was removed to concentrate on nickel (line 214) .
In Line 258, the acronym PGP should be changed for plant growth promoting, since was not described previously.
We have specified the meaning of the acronym PGP in the manuscript.
Since the root is being studied, Ni concentration levels of the root should have been measured, giving more strength to the ideas of: “highlight a Ni translocation- Line 216”, “do not show differences at increasing Ni - Line 230”, “T. arvense, specifically at the root level - Line 250” and “immobilized at the root level - Line 433”.
Thank you for the comments. We have given more emphasis to the sentences on the basis on our data, so we have rearranged the sentences as indicated below:
"…DMG test highlight an evident Ni translocation to aboveground biomass…" – Line 233-235
"The roots systems of both hyperaccumulators A. utriculata and N. caerulescens do not show significant differences in terms of roots surface and biomass at increasing Ni" – Line 248-249
"... T. arvense, specifically at the root level where root surface and biomass increase to cope with high level of Ni" – Line 269-272
"... metal is immobilized at the root level to avoid translocation to shoot biomass and possible related toxicity symptoms" – Line 453-455
Reviewer 3 Report
Dear Editors,
Thank you so much for choosing me as a reviewer of the manuscript ID plants-1122285 entitled “Root response to nickel in hyperaccumulator and non-hyperaccumulator species”.
Detailed remarks concerning manuscript ID plants-1122285.
The clear scientific hypothesis and the purpose of the reports should be given. The clear conclusions with the answer to the question stated as a scientific hypothesis together with the directions for the future studies also should be given.
All the figures and tables should be clear for the reader without need to refer to the text of the manuscript. Please do needed changes adding needed explanations.
“Each plant was gently removed from the substrate, washed with tap water and then with deionised water, divided into root and shoot and weighed for fresh biomass. Roots were scanned and the resulted images were processed and analysed as in paragraph 3.2.2 to assess the root surface area” I could not find in the manuscript the paragraph 3.2.2. The clear assay concerning the root surface area analysis should be provided. Section 3 is the ‘Discussion’. The discussion section should be divided into subsection similar to the subsections from ‘Results’ section. The ‘discussion’ section should be expanded and each of the analyzed parameter group should be discussed in the separate subsection.
Reference list
Reference list should be checked. It should be prepared strictly according to the guides for authors. There are many editorial mistakes in it. It is impossible to mention all of them. For example once the journal tiles are abbreviated but the other time not. All the Latin names of species should be italicized. Once each word of the journal tile is written with capital letter but the other time not. Please go through the whole reference list check it very carefully and do needed changes.
“Trotta, A.; Falaschi, P.; Cornara, L.; Minganti, V.; Fusconi, A.; Drava, G.; Berta, G. Arbuscular Mycorrhizae Increase 588 the Arsenic Translocation Factor in the As Hyperaccumulating Fern Pteris Vittata L. Chemosphere 2006, 65, 74–81, 589 doi:10.1016/j.chemosphere.2006.02.048” why “as” is written with capital letter?

Author Response
Detailed remarks concerning manuscript ID plants-1122285.
The clear scientific hypothesis and the purpose of the reports should be given. The clear conclusions with the answer to the question stated as a scientific hypothesis together with the directions for the future studies also should be given.
Thank you for the useful comments and suggestions.
To better explain the purpose of the manuscript, we added a scientific hypothesis in the Introduction as research question, whose answer was provided in the conclusions along with practical implications. Instead, the future research perspectives had already been indicated at the end of the Discussion section.
All the figures and tables should be clear for the reader without need to refer to the text of the manuscript. Please do needed changes adding needed explanations.
As suggested, we have provided the necessary explanations in figure and table captions as below:
Table 1: we have specified which biological parameters are considered and we have changed "Total Area" to "Root surface area" and we have removed the unit of measurement of the Ni treatment
Table 2: we have indicated the unit of measurement of the Ni treatment
Figure 4: we specified the seven parameters evaluated in the transient fluorescence analysis.
We also removed the wording "Pot test" from the captions of the three tables.
“Each plant was gently removed from the substrate, washed with tap water and then with deionised water, divided into root and shoot and weighed for fresh biomass. Roots were scanned and the resulted images were processed and analysed as in paragraph 3.2.2 to assess the root surface area” I could not find in the manuscript the paragraph 3.2.2.
Thank you for the observation. It was a mistake, so we have deleted the reference to the paragraph 3.2.2.
The clear assay concerning the root surface area analysis should be provided.
In paragraph 2.1 we reported the qualitative and quantitative analysis of the root area of the test species performed with the ImageJ software.
Qualitative analysis is supported by root system images (S1-S4) scanned and processed at different Ni concentrations and published online as Supplementary Materials.
On the other hand, the quantitative analysis is represented in Figure 1 and commented in this same paragraph 2.1
We hope that this could be considered exhaustive by the reviewer.
Section 3 is the ‘Discussion’. The discussion section should be divided into subsection similar to the subsections from ‘Results’ section. The ‘discussion’ section should be expanded and each of the analyzed parameter group should be discussed in the separate subsection.
We reorganized the discussion as the results also removing the reference to the "Pot test" paragraph. We have kept only the subdivision between "Evaluation of root area, biomass, and plant water content" and "Photosynthetic efficiency". In this way it was more evident the detailed discussion of the subsections, as suggested.
Reference list
Reference list should be checked. It should be prepared strictly according to the guides for authors. There are many editorial mistakes in it. It is impossible to mention all of them. For example once the journal tiles are abbreviated but the other time not. All the Latin names of species should be italicized. Once each word of the journal tile is written with capital letter but the other time not. Please go through the whole reference list check it very carefully and do needed changes.
As recommended in the guidelines of the Instructions for the authors of Plants we used a bibliography software (Zotero). Sometimes there may be some inaccuracy in the citation given by the software (i.e. incomplete abbreviations, Latin name italicized, capital letters). Although we have carried out a whole check, we apologize for any errors and omissions that may have been overlooked.
We have also added the DOI number (Digital Object Identifier) in the citations and abbreviated the names of the scientific journals one by one.
All scientific names of plants were italicized.
“Trotta, A.; Falaschi, P.; Cornara, L.; Minganti, V.; Fusconi, A.; Drava, G.; Berta, G. Arbuscular Mycorrhizae Increase the Arsenic Translocation Factor in the As Hyperaccumulating Fern Pteris Vittata L. Chemosphere 2006, 65, 74–81, 589 doi:10.1016/j.chemosphere.2006.02.048” why “as” is written with capital letter?
In this case “As” is the atomic symbol of chemical element Arsenic and requires the capitalization of the first letter.